

# Spatiotemporal dynamics of land use land cover change and its drivers in the western part of Lake Abaya, Ethiopia

Zeleke Assefa Getaneh[1,2], Sebsebe Demissew[1] and Zerihun Woldu[1]

[1] Department of Plant Biology and Biodiversity Management, Addis Ababa University, Addis Ababa, Ethiopia
[2] Department of Biology, Arba Minch University, Arba Minch, Ethiopia

## ABSTRACT

Understanding the dynamics of land use/land cover (LU/LC) changes and what drives these changes is essential for creating effective strategies for sustainable land management. It also helps to monitor the impact on ecosystems and biodiversity, which is crucial for policy-making. This study focused on assessing the trends, rates, and extent of LU/LC change and its causes in the western part of Lake Abaya in Ethiopia. To achieve this, we used a supervised classification method with a maximum-likelihood algorithm to map different land use land cover types. Additionally, we gathered information through field observations, focus group discussions (FGDs), and key informant interviews (KIIs) to identify the factors driving LU/LC change and its consequences between 1990 and 2022. The study findings revealed that vegetation and wetlands significantly decreased over this period, while water bodies, agricultural land, and settlements expanded at the expense of other land uses. The average normalized difference vegetation index (NDVI) values decreased from 0.368 in 1990 to 0.135 in 2022, indicating declining vegetation health. Local communities point to several factors responsible for these changes, including the expansion of agricultural land, increased settlement, firewood collection, and charcoal production (as proximate/immediate drivers), as well as population growth, poverty, unemployment, climate change, and policy-related issues (as underlying causes). Thus, it needs the development and implementation of an integrated and sustainable land management system, and strong land use and restoration policies in order to halt or reduce the rapid expansion of agricultural land and settlement areas at the expense of vegetation and wetlands.

Corresponding author
Zeleke Assefa Getaneh,
zelekepg@gmail.com

## INTRODUCTION

Land use/land cover change represent the dynamic alterations occurring on Earth's terrestrial surface, driven by a range of natural and human-related factors. This process is intricate and influenced by various aspects, including biophysical, socio-economic, and economic factors (*Arsanjani, 2012*). While it has been occurring for centuries, the pace of change has accelerated in recent decades, largely due to human activities. These alterations can have far-reaching consequences for the environment, including local and global climate impacts, loss of biodiversity, and changes in ecosystem services (*Pereira,*

*2020*). Understanding these transformations and their environmental effects is crucial for developing effective strategies to ensure a sustainable future.

The term "land use" pertains to human activities directly related to the utilization of land, while "land cover" encompasses the natural vegetation and artificial structures that overlay the land's surface (*Anderson et al., 1976*). In other words, land use describes human actions on land, while land cover refers to the types of vegetation covering a specific area (*Sherbinin, 2002*). According to *Campbell & Wynne (2011)*, land cover encompasses the visible features on the Earth's surface, including both natural elements and those modified by human influence.

Presently, at various scales (local, regional, and global), changes in land use/land cover are causing unprecedented disruptions in ecosystems and environmental processes, resulting in significant impacts such as increased natural disasters, loss of biodiversity, altered ecosystem services, forest fragmentation, and disruptions to socio-cultural practices (*Kindu et al., 2013*). As highlighted by *Alemu et al. (2015)*, the current rate, extent, and intensity of land use/land cover change far exceed anything seen in human history and are the leading causes of major environmental concerns, including climate change, biodiversity loss, natural disasters, and pollution of water, soil, and air.

*Geist & Lambin (2002)* pointed out that both natural and human-induced (anthropogenic) forces contribute to land use/land cover change, with anthropogenic forces playing a dominant role. While natural forces such as continental drift, glaciations, flooding, and tsunamis have historically led to land use/land cover changes (*Yesuph & Dagnew, 2019*), the recent dynamics are primarily driven by human activities, including the conversion of forested areas into agricultural land, urban expansion, and settlement growth (*Giri, 2012*). This unprecedented rate of change, which is leading to substantial alterations in nearly all ecosystems around the world, has now become a significant global environmental concern.

Human interactions with nature to meet their needs have transformed the Earth's surface in ways unmatched by any other species or factors (*Melaku, 2016*). These human-induced forces can be categorized into two main groups: proximate/immediate drivers and indirect forces (*Geist & Lambin, 2002*). Proximate drivers encompass direct human actions on local land cover, including the expansion of agriculture, unsustainable forest resource exploitation, and infrastructure development. Indirect forces, on the other hand, encompass economic, institutional, technological, cultural, and demographic changes that amplify the effects of proximate drivers on natural resource use (*Geist et al., 2006*).

Ethiopia, a country characterized by diverse vegetation zones, has also experienced the deterioration and depletion of natural vegetation resources, including forests, shrublands, woodlands, and grasslands in various regions. This has primarily resulted from land use/land cover changes driven by the high demand for agricultural land to support the rapidly growing population and the establishment of new settlements (*Alemu et al., 2015*; *Bessie et al., 2016*). The situation is particularly severe in the rift valley lake regions of the country due to the expansion of agriculture and settlement areas. In light of these challenges, this study was conducted in this area with the objectives of (i) assessing the trends, rates, and extents of land use/land cover change during the period 1990–2022,
(ii) identifying the primary land use types in the area, and (iii) proposing conservation strategies based on the study's findings.

# MATERIALS AND METHODS

## Study area description

This study was conducted in the western escarpment of Lake Abaya (the largest lake in the Great Rift Valley), which is located in the Gamo zone, southern Ethiopia. The specific study site is located in Mirab Abaya district, approximately 50 km to the north of Arba Minch. The district encompasses elevations ranging from 1,100 to 2,900 m above sea level and experiences an average annual rainfall of 800–1,600 mm and temperatures ranging from 24 to 30 °C.

The study area is precisely situated within coordinates 6°18′00″–6°39′00″N latitude and 37°35′05″–37°50′00″E longitude. It lies along the western edge of the southern segment of the Rift Valley in the South Ethiopia regional state (Fig. 1).

## Data acquisition (1990–2022)

We employed a combination of primary and secondary data sources to analyze the changes in land use/land cover (LU/LC) over time and space in our study region. These data were gathered from various sources. Ground control points (GCP) for ground truth verification were collected through field surveys using Garmin GPS devices. Additionally, we utilized Landsat images with high resolution and minimal cloud cover (less than 10%) for the years 1990, 2006, and 2022, which were accessed through the United States Geological Survey (USGS) Earth Explorer platform (https://earthexplorer.usgs.gov/). Specifically, Landsat images from Thematic Mapper (TM) for 1990, Enhanced Thematic Mapper Plus (ETM+) for 2006, and Operational Land Image (OLI) for 2022 were utilized in our analysis (Table 1). To ensure data consistency, all images were acquired during the dry season in the study area, typically in February and March, as these periods tend to have fewer cloud cover interruptions in the study area.

Furthermore, we incorporated secondary data, including shapefiles and topographic maps, which were obtained from the Ethiopian Mapping Agency (EMA).

## Image pre-processing and classification

Preparing satellite imagery before conducting image classification is crucial to minimize or eliminate errors caused by the instruments, rectify distorted data, and create a more accurate representation of the original scene. To achieve this, we followed a systematic process. Initially, the satellite images for each band in the years 1990, 2006, and 2022 were assembled in ERDAS 2015, resulting in a consolidated satellite image stack. Subsequently, we isolated the study area image from this stacked satellite image by using the Area of Interest (AOI) layer from the study area's shapefile.

For our classification approach, we employed a supervised classification technique utilizing a maximum-likelihood algorithm, which was guided by ground reference points. Prior to conducting this supervised classification, we conducted field observations to gather Ground Control Points (GCPs) for accurately georeferencing the images. This step helped
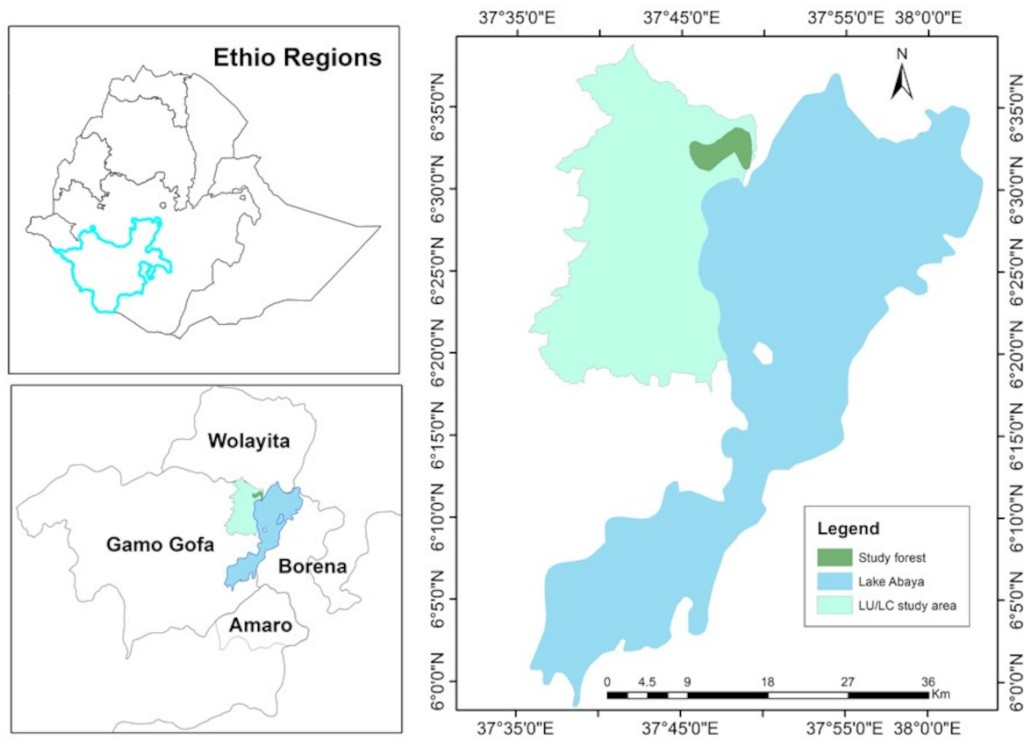

**Figure 1** **Location map of the study area.** Shape file data source: Ethiopian Mapping Agency, https://africaopendata.org/dataset/ethiopia-shapefiles.

**Table 1** **List of Landsat images used for LU/LC classification.**

| Acquisition date | Sensor type | Path and row | Spatial resolution | Source |
|---|---|---|---|---|
| 08/02/1990 | TM | 169/055 | 30 m | USGS |
| 27/01/2006 | ETM+ | 169/055 | 30 m | USGS |
| 15/03/2022 | OLI | 169/055 | 30 m | USGS |

us comprehend the characteristics of various land cover classes and supported the visual interpretation of the images. Additionally, we utilized Google Earth to acquire reference points for georeferencing purposes. These reference points were strategically selected ($\geq$50) based on the extent of each land cover class (*Congalton, 1991*), and their coordinates were recorded using GPS devices with an accuracy of approximately $\pm$3 m. We also verified these reference points on Google Earth to ensure their accuracy. The image classification process was executed using ArcGIS version 10.8. As a result of this supervised classification, we generated six distinct land use classes within the study area (Table 2).

## Accuracy assessment

We evaluated the accuracy of our classification process by employing two key methods: overall accuracy and Kappa statistics analysis. This evaluation was performed through the generation of a confusion matrix. To ensure the accuracy of the supervised classification

**Table 2  Land use land cover classes of the study area.**

| No. | LU/LC classes | Descriptions |
|---|---|---|
| 1 | Vegetation | Areas dominated with trees forming closed or nearly closed canopies, in which trees higher than 5 m and a canopy cover of more than 10% (Forests), areas covered with small trees, in which shrubs are the dominant vegetation (shrub land) and areas with sparse cover of trees (woodland). |
| 2 | Rain-fed crop land | Mosaic of small holder farms dominantly used for crop cultivation. |
| 3 | Irrigated crop land | Small-scale irrigated commercial farms along Lake Abaya used for crop cultivation/fruit and vegetable production. |
| 4 | Settlements | Areas or villages associated with buildings and towns. |
| 5 | Wetlands | Areas of land near lake Abaya saturated with water which support both aquatic and terrestrial species. |
| 6 | Water body | Areas covered by surface water /lake/. |

results for land use/land cover (LU/LC) changes, we cross-referenced them with ground truth data. The ground control points (GCPs) established during fieldwork, as well as information from Google Earth images, were used for this validation process.

For the years 1990 and 2006, reference points were collected from corresponding Google Earth images. In the case of 2022, reference points were gathered through a combination of field observations and Google Earth imagery. To assess the accuracy of the LU/LC classification results, we utilized ERDAS Imagine 2015 software to determine how closely they aligned with the actual conditions on the ground. We created a confusion or error matrix, which encompasses both overall accuracy and the Kappa coefficient, based on the supervised LU/LC maps for the study periods.

The overall accuracy was computed by dividing the total number of correct classifications (sum of the main diagonal in the confusion matrix) by the total number of pixels in the matrix, following the methodology outlined by *Park et al.* in *2013*. Additionally, Kappa statistics were calculated using Eq. (1), as described by *Lillesand, Kiefer & Chipman* in *2015*, relying on the GCPs collected from the identified land use types. This comprehensive assessment helped us gauge the accuracy of our LU/LC classification results and their consistency with real-world conditions.

$$K = \frac{N \sum_{i=1}^{r} x_{ii} - \sum_{i=1}^{r}\left(x_{ij} * x_{ji}\right)}{N^2 - \sum_{i=1}^{r}\left(x_{ij} * x_{ji}\right)} \tag{1}$$

where $K$ is the Kappa coefficient, $N$ is the total number of samples in the matrix, r corresponds to the number of rows in the matrix, $x_{ii}$ is the number in row $i$ and column $i$, $x_{ji}$ is the total for row $i$, and $x_{ij}$ is the total for column $i$ and $^\star$ denotes multiplication.

## Land use/land cover change detection

We employed a change detection technique known as post-classification comparison to identify land use land cover (LU/LC) changes. This involved the use of separately classified Landsat images, followed by a comparison of the resulting LU/LC maps. The choice of this

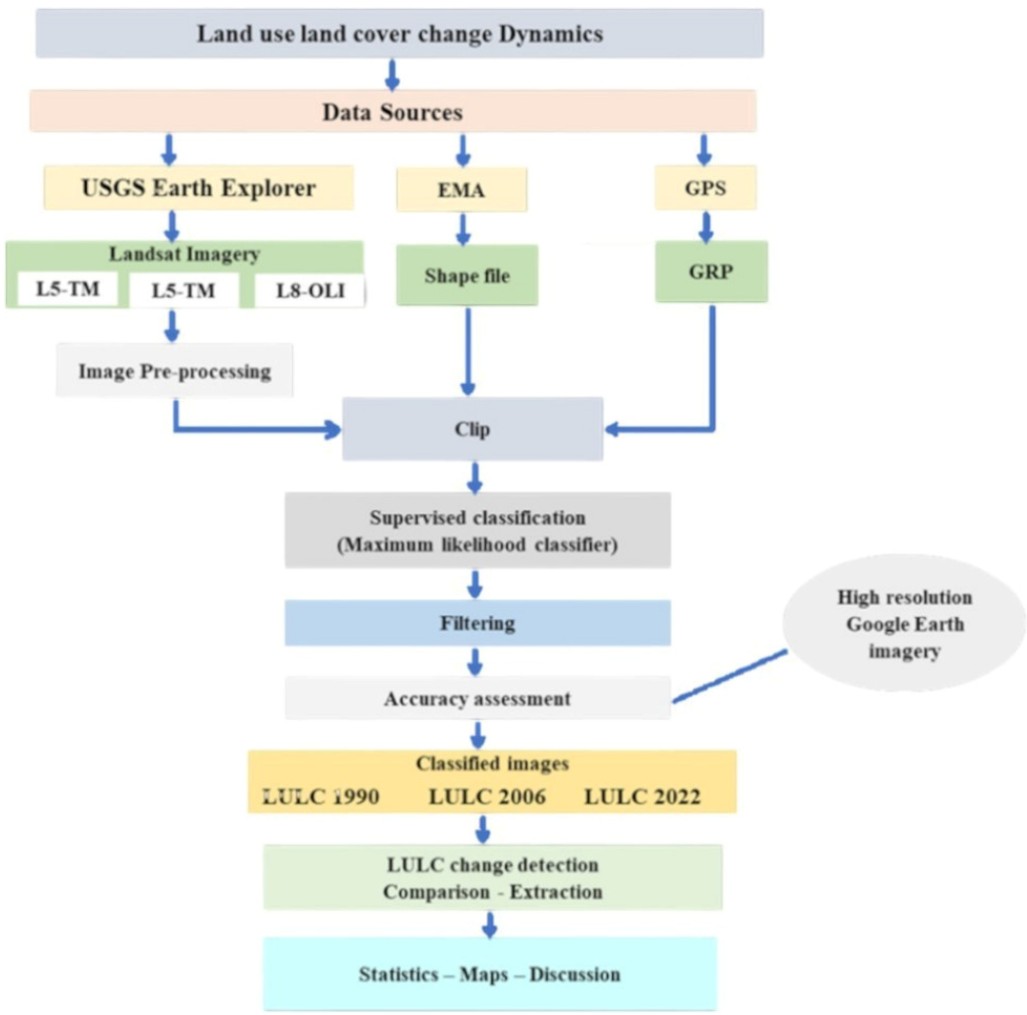

**Figure 2  A flowchart of the LU/LC change study.**

method was driven by its widespread use and effectiveness in LU/LC change detection, as supported by previous studies (*Foody, 2002*; *Teferi et al., 2013*).

 Subsequently, we calculated the percentage and rate of change for each LU/LC class using Eqs. (2) and (3). The entire process, from data acquisition to change detection, adhered to a structured methodology outlined in Fig. 2.

$$PoC = \frac{A2 - A1}{A1} \times 100 \tag{2}$$

$$RoC = \frac{A2 - A1}{A1} \times \frac{1}{(T2 - T1)} \times 100 \tag{3}$$

 where RoC = rate of change, PoC = percentage of change of a particular LULC class, A1 = area of the previous land use class, A2 = area of the recent land use class and $T2$ = current year, and $T1$ = previous year.

### Normalized Difference Vegetation Index (NDVI)

We employed the Normalized Difference Vegetation Index (NDVI) as a vegetation indicator to assess the extent of vegetation coverage in the study area. NDVI is a commonly utilized metric for gauging the health and density of vegetation using sensor data. It yields excellent outcomes in the analysis of multispectral remote sensing images, particularly for areas with varying vegetation densities and dispersed vegetation types, as substantiated by *Gandhi et al.* in *2015*. The calculation of NDVI is shown in Eq. (4).

$$NDVI = \frac{(NIR - R)}{(NIR + R)} \tag{4}$$

where: NDVI = Normalized difference vegetation index, NIR = Near infra-red, R = Red

### Major drivers of LU/LC changes in the study area

We gathered information regarding the primary factors influencing changes in land use/land cover (LU/LC) and the perspectives of local communities on these alterations. This data was collected through key informant interviews (KIIs), focus group discussions (FGDs), and on-site observations. To achieve this, a set of questions was prepared to evaluate the views of local residents regarding the trends, patterns, and drivers of changes in land use/ land cover. In total, we conducted 22 FGDs, each involving 5–7 individuals from various social groups, and 40 KIIs with individuals who have resided in the area for over 40 years. These activities aimed to obtain qualitative insights into LU/LC changes in the study area. The selection of participants for the FGDs and KIIs was purposeful, with input from community elders and agricultural extension workers from 20 different kebeles, as they were expected to own knowledge about LU/LC modifications and their underlying causes.

## RESULTS

### Land use land cover classification

We identified five major land use land cover classes in the study site. Namely, water bodies, wetlands, vegetation, settlements, rain-fed cropland and irrigated cropland for the years 1990, 2006 and 2022 (Fig. 3). The findings also indicated that the combined land area covered by vegetation and wetlands exhibited a decreasing pattern over the three consecutive times. Conversely, the areas of water bodies, croplands (both rain-fed and irrigated), and settlements (residential areas) displayed an increasing trend, as summarized in Table 3.

### Accuracy assessment

The error/ confusion matrix analysis results revealed that the overall accuracy of the study site for the years 1990, 2006 and 2022 was 90%, 93% and 94%, respectively, with a corresponding Kappa coefficient value of 87%, 91% and 91% (Tables 4, 5 and 6).

### Land use land cover change dynamics

The findings of the study indicated that certain land uses exhibited an upward trajectory, while others displayed a declining pattern (Table 7). Nonetheless, on the whole, the results

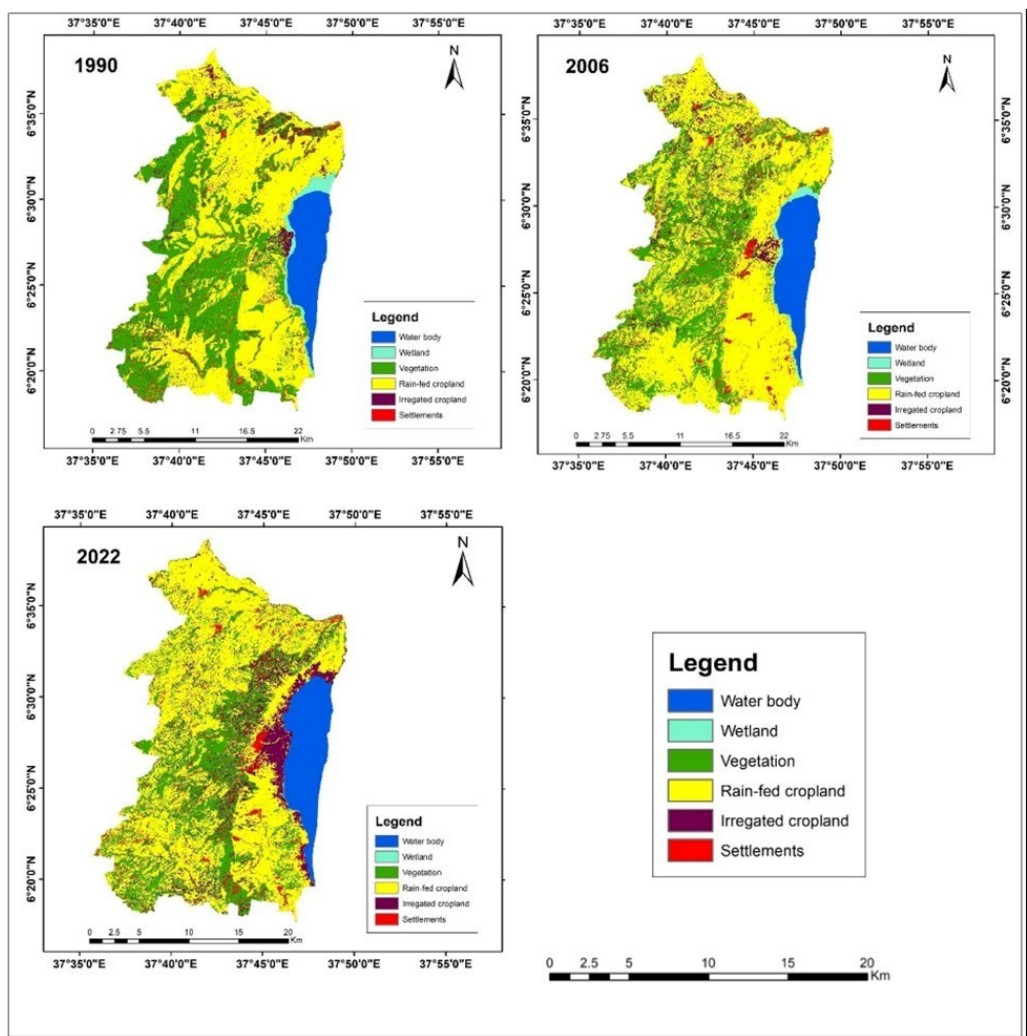

**Figure 3** **Land use/land cover class maps of the study area for the years 1990, 2006 and 2022.** Shape file data source: Ethiopian Mapping Agency, https://africaopendata.org/dataset/ethiopia-shapefiles.

indicated a persistent and substantial transformation in land use/land cover within the research area. To illustrate, vegetation and wetlands experienced significant reductions of 47.4% and 94.2%, respectively. Specifically, vegetation cover decreased from 22,960.7 hectares (37.6%) in 1990 to 12,078 hectares (19.8%) in 2022, while the wetland area decreased from 1,034.8 hectares (1.7%) in 1990 to 60.0 hectares (0.1%) in 2022. Conversely, agricultural land (both rain-fed and irrigated), settlements, and water bodies displayed an upward trend. Agricultural land increased from 30,004.5 hectares, settlements from 2,198.9 hectares, and water bodies from 4,934.3 hectares in 1990 to 40,754 hectares, 2,502 hectares, and 5,739 hectares, respectively, in 2022, with respective percentage changes of 167.4%, 13.8%, and 16.3%.

The analysis of land use/land cover changes between 1990 and 2022, as shown in Table 8, highlights that the primary focus of conversion has been on vegetation. Specifically,

**Table 3   Land use land cover classes and their area coverage in hectare and percentage.**

| Land use type | Periods/Years | | | | | |
|---|---|---|---|---|---|---|
| | 1990 | | 2006 | | 2022 | |
| | Area (ha) | % | Area (ha) | % | Area (ha) | % |
| Water body | 4,934.3 | 8.1 | 5,136 | 8.4 | 5,739 | 9.4 |
| Wetland | 1,034.8 | 1.7 | 652 | 1.1 | 60 | 0.1 |
| Vegetation | 22,960.7 | 37.6 | 18,881 | 30.9 | 12,078 | 19.8 |
| Rain-fed Cropland | 29,413.4 | 48.1 | 31,008 | 50.7 | 33,702 | 55.1 |
| Irrigated cropland | 591.1 | 1.0 | 3,097 | 5.1 | 7,052 | 11.5 |
| Settlements | 2,198.7 | 3.6 | 2,359 | 3.9 | 2,502 | 4.1 |
| Total | 61,133 | 100 | 61,133 | 100 | 61,133 | 100 |

**Table 4   Error/confusion matrix of the year 1990.**

| | Water body | Wetland | Vegetation | Rain-fed cropland | Irrigated cropland | Settlements | Row total |
|---|---|---|---|---|---|---|---|
| Water body | **83** | 7 | 0 | 0 | 2 | 0 | 92 |
| Wetland | 4 | **130** | 2 | 5 | 5 | 0 | 146 |
| Vegetation | 2 | 6 | **436** | 16 | 12 | 0 | 472 |
| Rain-fed Cropland | 1 | 4 | 14 | **266** | 15 | 0 | 300 |
| Irrigated Cropland | 0 | 2 | 5 | 7 | **109** | 0 | 123 |
| Settlements | 0 | 0 | 7 | 5 | 2 | **96** | 110 |
| Column total | 90 | 149 | 464 | 299 | 145 | 96 | 1,243 |
| User's Accuracy | 90.2 | 89.0 | 92.4 | 88.7 | 88.6 | 87.3 | |

Overall accuracy = 90.1%, Kappa coefficient = 87%

**Table 5   Error/confusion matrix of the year 2006.**

| | Water body | Wetland | Vegetation | Rain-fed cropland | Irrigated cropland | Settlements | Row total |
|---|---|---|---|---|---|---|---|
| Water body | **31** | 1 | 1 | 0 | 0 | 0 | 33 |
| Wetland | 0 | **48** | 0 | 1 | 2 | 0 | 51 |
| Vegetation | 0 | 1 | **238** | 5 | 6 | 0 | 250 |
| Rain-fed Cropland | 0 | 5 | 12 | **344** | 10 | 3 | 374 |
| Irrigated Cropland | 1 | 2 | 0 | 4 | **50** | 0 | 57 |
| Settlements | 0 | 0 | 2 | 4 | 1 | **133** | 140 |
| Column total | 32 | 57 | 253 | 358 | 69 | 136 | 905 |
| User's Accuracy | 93.9 | 94.1 | 95.2 | 92.0 | 87.7 | 95.0 | |

Overall accuracy = 93.3%, Kappa coefficient = 91%

vegetation has predominantly been transformed into two categories: crop land, which includes both irrigated (covering 162.5 hectares) and rain-fed (covering 6,702.8 hectares) areas, totaling 6,865.3 hectares, and settlements (covering 12.5 hectares). This underscores the rapid expansion of crop land and settlement areas, with these land uses expanding

**Table 6  Error/confusion matrix of the year 2022.**

|  | Water body | Wetland | Vegetation | Rain-fed cropland | Irrigated cropland | Settlements | Row total |
|---|---|---|---|---|---|---|---|
| Water body | **22** | 3 | 0 | 0 | 0 | 0 | 25 |
| Wetland | 0 | **38** | 0 | 2 | 1 | 0 | 41 |
| Vegetation | 0 | 2 | **290** | 6 | 5 | 0 | 303 |
| Rain-fed Cropland | 1 | 3 | 8 | **509** | 12 | 5 | 538 |
| Irrigated Cropland | 0 | 0 | 2 | 0 | **26** | 0 | 28 |
| Settlements | 1 | 2 | 4 | 5 | 2 | **130** | 144 |
| Column total | 24 | 48 | 304 | 522 | 46 | 135 | 1,079 |
| User's Accuracy | 88.0 | 92.7 | 95.7 | 94.6 | 92.9 | 90.3 | |

Overall accuracy = 94.1%, Kappa coefficient = 91%

**Table 7  Land use land cover change dynamics from the year 1990–2022.**

| Land use type | 1990–2006 | | | 2006–2022 | | | 1990–2022 | | |
|---|---|---|---|---|---|---|---|---|---|
|  | Change (ha) | % of change | Rate of change | Change (ha) | % of change | Rate of change | Change (ha) | % of change | Rate of change |
| Water body | 202 | 4.1 | 0.26 | 603 | 12.2 | 0.73 | 805 | 16.3 | 1.02 |
| Wetland | −383 | −37 | −2.31 | −592 | −57.1 | −5.67 | −975 | −94.2 | −5.89 |
| Vegetation | −1,980 | −22.4 | −0.59 | −6,803 | −6.8 | −2.25 | −8,783 | −29.2 | −2.63 |
| Rain-fed Cropland | 1,595 | 14.6 | 0.34 | 2,694 | −9.2 | 0.54 | 4,289 | 5.4 | 0.91 |
| Irrigated cropland | 406 | 15.1 | 0.94 | 3,955 | 147 | 7.98 | 4,361 | 162 | 10.13 |
| Settlement | 160 | 7.3 | 0.46 | 143 | 6.5 | 0.38 | 303 | 13.8 | 0.86 |

**Table 8  Land use land cover change matrix from 1990 to 2022 (in hectare).**

| | LULC class | Area of LU/LC cover classes in 1990 | | | | | | |
|---|---|---|---|---|---|---|---|---|
| | | Water body | Wetland | Vegetation | Rain-fed Cropland | Irrigated cropland | Settlements | Total |
| Area of LU/LC classes in 2022 | Waterbody | 5,739 | 0.0 | 0.0 | 0.0 | 0.0 | 0.0 | 5,739.0 |
| | Wetland | 4.7 | 47.5 | 0.0 | 0.0 | 7.8 | 0.0 | 60.0 |
| | Vegetation | 0.0 | 2.2 | 5,198.0 | 6,702.8 | 162.5 | 12.5 | 12,078.0 |
| | Rain-fed Cropland | 0.0 | 0.0 | 0.0 | 23,766.5 | 9,449.3 | 486.2 | 33,702.0 |
| | Irrigated cropland | 0.0 | 316.1 | 0.0 | 0.0 | 6,735.9 | 0.0 | 7,052.0 |
| | Settlements | 0.0 | 0.3 | 0.0 | 0.3 | 0.0 | 2,501.4 | 2,502.0 |
| | Total | 5,743.7 | 366.1 | 5,198.0 | 30,469.6 | 16,355.5 | 3,000.1 | 61,133.0 |

significantly at the expense of other types of land use, notably vegetation. Furthermore, the analysis results also revealed the conversion of wetlands into crop lands.

### Normalized difference vegetation index (NDVI) changes

NDVI is used to quantify vegetation greenness and is a measure of vegetation health. It is used in understanding vegetation density and to evaluate the specific area's vegetation

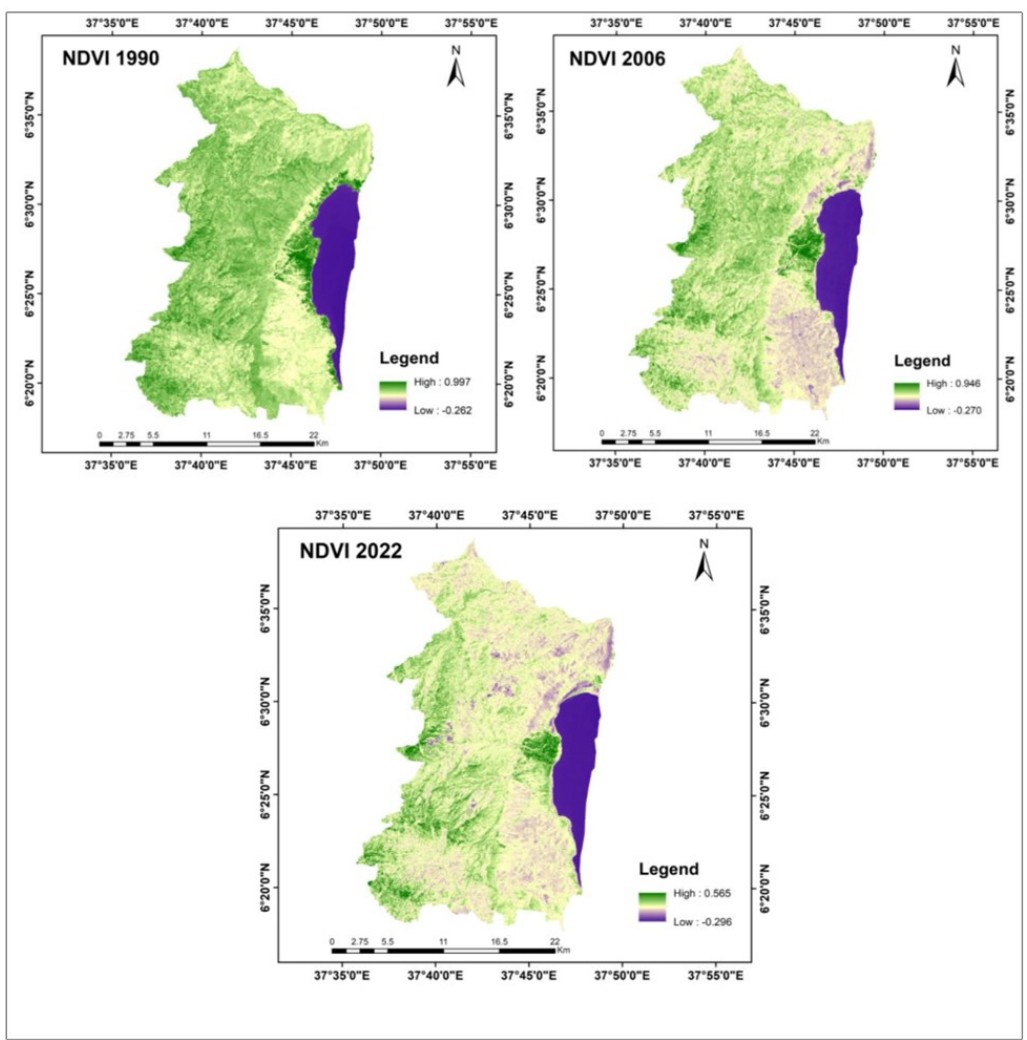

**Figure 4** **The NDVI maps of the study area for the years 1990, 2006 and 2022.** Shape file data source: Ethiopian Mapping Agency, https://africaopendata.org/dataset/ethiopia-shapefiles.

health (*Hussain & Karuppannan, 2023*). We derived three NDVI images for the years 1990, 2006 and 2022 using the corresponding Landsat images of each year (Fig. 4). The NDVI values ranged between −0.262 and +0.997 for the year 1990, −0.270 and +0.946 for the year 2006 and −0.296 and +0.565 for the year 2022 (Table 9). The NDVI maps showed a successive decrease in the vegetation cover of the study area, revealing the existence of continuous land use land cover changes in the studied periods.

## Major drivers of LULC changes in the study area
### Proximate drivers
According to the findings gathered from discussions and interviews with community members in the study area, the proximate/immediate factors driving changes in land use/ land cover include the expansion of agricultural land, the growth of settlement areas,

**Table 9   NDVI values of the study area for the years 1990, 2006 and 2022.**

| Years | High value | Low value | Mean | Sd |
|-------|-----------|-----------|------|-----|
| 1990 | +0.997 | −0.262 | 0.368 | 0.890 |
| 2006 | +0.946 | −0.270 | 0.338 | 0.860 |
| 2022 | +0.565 | −0.296 | 0.135 | 0.609 |

charcoal production, gathering firewood, tree cutting for various purposes, and overgrazing. Among these factors, the expansion of agricultural land and the growth of settlement areas were identified as the primary drivers of changes in land use/land cover, ranking first and second, respectively, followed by charcoal production, firewood collection, tree cutting for various purposes, and overgrazing as shown in Table S10.

### Underlying causes of LULC changes

We obtained four major underlying causes of LU/LC change in the study area from the local communities through the FGD and KIIs. Namely, population pressure, poverty (economic crises), climate change (precipitation variability), and policy and institutional factors, with population pressure being the leading cause followed by poverty (Table S11).

## DISCUSSION

### Evaluation of land use/land cover classification accuracy

The accuracy of classifying land use/ land cover in this study for the years 1990, 2006, and 2022 was impressive, with accuracy rates of 90%, 93%, and 94%, respectively. These rates exceed the widely accepted standard of 85% accuracy (as proposed by Anderson et al. in 1976), indicating strong agreement between the classified images and the actual ground truth. Additionally, the Kappa coefficient values for the same years were 87%, 91%, and 91%, further confirming the robust agreement between the classified images and the real land use data (as noted by Lea & Curtis in 2010 and McHugh in 2012.

### Changing patterns in land use and land cover

The findings of the study highlight significant and continuous changes in land use/ land cover over the 32-year period from 1990 to 2022. Notably, cultivated (agricultural) land and settlement areas exhibited substantial growth at the expense of other land use categories within the study area. This is in agreement with the findings of several studies where cultivated (agricultural) land and settlement (built up) areas are expanding in rift valley regions (Tsegaye et al., 2021; Mesfin et al., 2020; Dingamo et al., 2021; Deche et al., 2023) and across Ethiopia at the expense of other land use types (Gashaw et al., 2018; Regasa & Nones, 2022; Gedefaw, Denghua & Girma, 2023).

The vegetation cover in the study area exhibited a continuous decline, dropping from 37.6% in 1990 to 30.9% in 2006 and further to 19.8% in 2022. This decline can be attributed to the expansion of both irrigated and rain-fed agriculture as well as urban development in the study site. Wetlands followed a similar decreasing trend from 1990 to 2022, while water bodies, cropland (both irrigated and rain-fed), and urban areas experienced consistent growth. The expansion of agricultural and settlement areas at the expense of vegetation

emerged as the primary drivers behind the reduction in vegetation cover, largely due to the escalating demand for arable land to support the growing population and to accommodate residential development (*Dibaba, Demissie & Miegel, 2020*; Dingamo et al. 2022).

The study's results also revealed that the expansion of settlement areas to accommodate individuals displaced from the Gamo highlands due to natural disasters like landslides was the second main factor contributing to the decline in vegetation cover. Moreover, the conversion of wetlands into agricultural land showed a steady increase between 1990 and 2022, likely driven by the rise in small-scale commercial farming employing small-scale irrigation techniques using water from the lake for the production of fruits (such as mangoes and bananas) and vegetables (like tomatoes and onions) (*Zekarias et al., 2021*; *Dingamo et al., 2021*; *Zekarias & Gelaw, 2023*).

The water body showed an increasing trend in the study site, likely due to an increase in the surface area of lake Abaya. This could be attributed to various human activities in the lake's catchment areas, which led to reduced vegetation cover, increased surface runoff, soil erosion, nutrient losses from agricultural land, reduced soil fertility, and higher silt content and nutrient loads in the lake (*Awulachew & Horlacher, 2004*; *Gronewold et al., 2013*; *Ogato, Bantider & Geneletti, 2021*; *Tasgara & Kumar, 2022*; *Zekarias & Gelaw, 2023*). These activities have aggravated environmental issues in the catchment areas.

## Vegetation cover change analyzed through NDVI

The Normalized Difference Vegetation Index (NDVI) is a valuable tool for tracking the health and dynamics of vegetation, providing essential insights into how climate fluctuations impact ecosystems and agriculture. It quantifies vegetation health by assessing how plants reflect specific portions of the electromagnetic spectrum (*Choubin et al., 2019*). By measuring how plants absorb and reflect light at distinct frequencies, NDVI serves as a reliable indicator of vegetation density and growth conditions (*Faisal et al., 2020*; *Hu et al., 2023*). NDVI values consistently fall within the range of −1 to +1, with higher values signifying healthy vegetation and lower values indicating sparse or absent vegetation, such as in deserts, bare land, or water bodies (*Choubin et al., 2019*; *Pasternak & Pawluszek-Filipiak, 2022*).

Our analysis revealed a significant decline in NDVI values from 1990 to 2022. In 1990, the maximum NDVI value reached 0.997, indicating robust vegetation cover in the study area at that time. Conversely, by 2022, the NDVI value had dropped to 0.565, signaling pressure on vegetation cover and shifts to other land use types. Therefore, our NDVI analysis underscores the decrease in vegetation cover from 1990 to 2022, primarily driven by the expansion of agricultural land and urban areas within the study area. This reaffirms the pattern of successive changes in land use and land cover during this period, a trend consistent with findings in Ethiopia reported by various researchers (*Dingamo et al., 2021*; *Temesgen, Warkineh & Hailemicael, 2022*; *Worku et al., 2023*).

## Drivers of land use and land cover changes

Our research identified agricultural land expansion and settlement (urban) growth as immediate drivers, with population pressure and economic challenges (poverty and
unemployment) serving as underlying causes of land use/land cover changes within the study area, as determined through FGD and KIIs. Numerous studies have similarly pinpointed agricultural land expansion as a major catalyst for land use/ land cover changes (*Hailu, Mammo & Kidane, 2020*; *Bufebo & Elias, 2021*; *Winkler et al., 2021*; Dingamo et al., 2022). Settlement (urban) expansion, particularly in regions undergoing land reform, can significantly affect natural vegetation and watershed characteristics, leading to significant environmental repercussions (*Verma et al., 2020*). In Ethiopia, resettlement programs driven by the government have been identified as core drivers of land use/land cover changes, often lacking adequate consideration for natural resource management and sustainable resource utilization (*Bufebo & Elias, 2021*; *Temesgen, Warkineh & Hailemicael, 2022*).

Population pressure emerges as a dominant force propelling land use/land cover change. Numerous studies have highlighted demographic dynamics as the primary contributors to shifts in land cover (*Wubie, Assen & Nicolau, 2016*; *Gebrehiwot et al., 2020*; *Dibaba, Demissie & Miegel, 2020*; *Bufebo & Elias, 2021*). The escalating demand for cultivated and urban land significantly drives the transformation of natural vegetation resources into agricultural and urban landscapes (*Anteneh, 2022*; *Mariye, Jianhua & Maryo, 2022*). Therefore, population pressure and poverty stand out as pivotal factors behind these changes, amplified by the demand for fuelwood, charcoal production, overgrazing, climate change (resulting in rainfall variability), and policy and institutional influences –all of which were raised by respondents during FGD and KII sessions.

The demand for fuelwood and charcoal production remains prominent drivers of land use/land cover changes. Although large-scale deforestation may not occur in the short term, imbalances between demand and supply can lead to deforestation and forest degradation (*Butz, 2013*; *Hailu, Mammo & Kidane, 2020*). Tree removal for timber, fuelwood, construction, and charcoal production significantly contributes to forest cover decline, thereby driving land use/land cover changes (*Munthali et al., 2019*; *Anteneh, 2022*; Dingamo et al., 2022). Overgrazing also plays a role in these changes, potentially leading to soil erosion, desertification, and biodiversity loss, with lasting environmental consequences (*Turi, Hayicho & Kedir, 2019*; *Hailu, Mammo & Kidane, 2020*).

Moreover, poverty and unemployment were identified as underlying causes of land use and land cover changes in the study area. Limited alternative income opportunities often drive land conversion for activities such as agriculture, charcoal production, firewood collection, and timber harvesting, ultimately contributing to deforestation. These findings align with the research of *Munthali et al. (2019)* and *Temesgen, Warkineh & Hailemicael (2022)*, who identified poverty and unemployment as significant drivers of land use/land cover changes.

Lastly, policy and institutional factors exert considerable influence on land use/ land cover changes in the study area. These changes are multifaceted and dynamic, influenced by various factors operating across different spatial and temporal scales. They can create opportunities and constraints for new land uses, increasingly affected by global factors like markets and policies (*Lambin et al., 2001*). Numerous previous studies in Ethiopia have underscored the role of policy and institutional factors as primary drivers of land

use/land cover changes (*Abera, Yirgu & Uncha, 2020*; *Hailu, Mammo & Kidane, 2020*; *Ogato, Bantider & Geneletti, 2021*).

Additionally, climate change emerges as a key underlying factor influencing land use/land cover changes in the study area, impacting traditional agricultural practices. Prior to the introduction of irrigated commercial farms near Lake Abaya, the region had a sparse population, relying on traditional rain-fed agriculture without modern inputs (*e.g.,* hybrid seeds, inorganic fertilizers, pesticides). However, due to climate change-induced rainfall variability and drought, coupled with modern farming practices, the majority of the population shifted from cattle farming to crop production (such as vegetables, horticulture, and fruits) using irrigation from the lake and tributaries. This shift in farming practices, driven by climate change, may lead to an increase in agricultural land at the expense of vegetation cover, as also reported by *Deche et al. (2023)*. Climate change, with its alterations in temperature, precipitation, and other climate variables, significantly influences land suitability for various uses, prompting farmers to adapt by altering their crops, potentially resulting in increased agricultural land and land use/land cover changes.

## CONCLUSIONS

The primary objective of this study was to evaluate the scale and direction of land use/land cover (LU/LC) changes and their drivers in the western vicinity of lake Abaya. We accomplished this by analyzing multi-temporal remote sensing data alongside field observations, focus group discussions (FGDs), and key informant interviews. The findings revealed a significant and continuous changes in land use/land cover over the 32-year period from 1990 to 2022. The expansion of agricultural land (both rain-fed and irrigated) and settlement areas expansion showed substantial growth at the expense of other land use categories. The vegetative cover and wetlands experienced a continuous decline, primarily due to the expansion of agriculture and various human-induced factors.

It is crucial to recognize that the dynamics of LU/LC changes are intricate, multifaceted, and context-dependent, varying across regions and circumstances. Investigating in LU/LC change patterns is an indispensable element for comprehending the interplay between human activities and the environment. Therefore, monitoring and mapping these changes are vital for effective natural resource management and utilization. Knowledge about the current LU/LC patterns and changes over time is fundamental for assessing environmental conditions and providing input for sustainable natural resource management. Furthermore, understanding the spatial and temporal dynamics of LU/LC, its drivers, and associated implications for vegetation cover is imperative for safeguarding ecosystems, biodiversity, and formulating effective conservation strategies.

This study establishes a foundational understanding for relevant government and non-governmental entities. It offers insights into the direction, rate, and extent of LU/LC changes, as well as their drivers and environmental impacts. Consequently, it recommends the adoption of integrated and sustainable land management practices and the implementation of robust land use and restoration policies. These measures are imperative for curbing or mitigating the rapid expansion of agricultural and urban areas

at the expense of vegetation and wetlands, ultimately promoting responsible and balanced land use practices.

## ACKNOWLEDGEMENTS

We are thankful for the local communities for being volunteer to give the necessary information.

### Funding
The authors received no funding for this work.

### Competing Interests
The authors declare there are no competing interests.

### Author Contributions
- Zeleke Assefa Getaneh conceived and designed the experiments, performed the experiments, analyzed the data, prepared figures and/or tables, authored or reviewed drafts of the article, and approved the final draft.
- Sebsebe Demissew analyzed the data, prepared figures and/or tables, authored or reviewed drafts of the article, and approved the final draft.
- Zerihun Woldu analyzed the data, prepared figures and/or tables, authored or reviewed drafts of the article, and approved the final draft.

### Data Availability
   The raw measurements are available in the Supplemental Files.

### Supplemental Information
Supplemental information for this article can be found online at http://dx.doi.org/10.7717/peerj.17892#supplemental-information.

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
