# Peer review of "Spatiotemporal dynamics of land use land cover change and its drivers in the western part of Lake Abaya, Ethiopia"

_PeerJ, doi:10.7717/peerj.17892_

## Round 0.1 · original submission · Major Revisions

In this study, the authors used a supervised classification method with a maximum-likelihood algorithm to map different land use land cover types.
In general, this manuscript needs major revision before its publication. Please revise this manuscript based on these valuable comments and suggestions.

Reviewer 1 ·

Basic reporting

no comment

Experimental design

no comment

Validity of the findings

no comment

Additional comments

In the abstract section, I suggest that the author emphasize the necessity of this study. By the way, please concise the abstract.
 Now, there are lots of experts to study this research filed. However, could you tell us your paper’s novelty?
 In the methods section, the author employed a supervised classification technique utilizing a maximum-likelihood algorithm to study, but this method is very common. And I don’t know why do not adopt the other classification method, such as SVM, RF etc. Please explain it.
 I suggest the author can give us your paper limitations.
 The conclusion section requires further concise to enhance the clarity of the paper's findings and contributions. Additionally, it is important to establish a clear connection and distinction between the conclusion and abstract.
 The quality of the figure need improvement. Such as Figure 2, Figure 3, please check the whole paper.
To sum up, I believe that this manuscript could make a contribution to this research filed and suggest the author make a revision and resubmit it.

Reviewer 2 ·

Basic reporting

- The English used in writing the article is clear and unambiguous except in one or two cases in line 40, and 24, where the sentences needed review
- Authors have provided sufficient evidence in the field/background/context and literatures referenced, the intext-reference/s for the list of natural causes in line 75 should be provided.
- The work has been structured appropriately, figures are good, tables are good and ok, with the exception of table 10 which i am not sure is correct because data from the key informant and focus group discussions cannot be reported in one table. The authors need to separate the two reports, the data from the key informants can be reported in a table but that from the focus groups can also be reported in another table but not in figures but text form e.g. a good number of the focus group participants identified agriculture expansion as one of the key direct drivers of LU/LC changes.
- The research findings addressed some of the research questions, but it would have been better to test if there are any relationship between LU/LC changes and LU/LC drivers like rainfall and temperature in the study area.

Experimental design

- The research met the requirement of originality but requires some statistical tests to show relationship between LU/LC changes and their drivers
- The research questions are somehow well defined, relevant and meaningful, but however if there was another question on the relationship between the LU/LC changes and their drivers, especially the climatic drivers (temperature and rainfall) and that would have help explained why some LU/LC transformed drastically.
- The procedures in collecting and processing images are well explained and followed but i don't know why the authors did not use farmers as key informants, instead collected data from only extension workers and community leaders - line 204
- It would have also been to name the communities in which focus group discussion were done- lines 204-207

Validity of the findings

-The research provide some useful information on LU/LC transformation, especially the changes in the original land cover over time, however it would have been better if the data on the drivers of LU/LC collected from the two groups of participants were reported separately.
- It would have also been better to name the types of vegetation found in the study area

Additional comments

Spatiotemporal dynamics of land use land cover change and its drivers in the western part of Lake Abaya, Ethiopia. (#91710)
Comments
Abstract-
- Line 40, it is better to use proximate/immediate drivers

Introduction and background
- Line 75 (list of natural causes should be supported by citation(s)
- Line 83- better to use proximate/immediate drivers, instead of just immediate drivers,
- Line 96- used in this area or name the area, instead of the area.
Materials and methods
- Line 106- give the study area coordinates as shown in the figure. This should be written before describing the physical regions (highlands and lowland features) and then climate and vegetation.
- Line 120- few cloud cover interruptions, where? be specific
- Line 204, 40 KIIs is good but you should have included farmers who have been using the land instead of interviewing only community leaders and extension workers. Remember most extension workers might have not been in the study area by 1990 but older farmers would have.
- Line 204- 22 focus group discussions. This is a good number but if it is done in one community, it becomes too many. So, state the communities in the study area where FGDs were conducted and give us the number done in each community.
- The instrument or tool used for the KII and the FGDs should have been separated because questions posed to each category might not necessary be the same.

Results
Line 212- Table 2, are the description of the LU/LC based on what you think they are or from another source?
Line 213- vegetation, which types specifically e.g., mangrove, grass etc.
Line 228- from 1,038.4 to 60 for changes in wetland between 1990-2022, what is the percentage change? It is important to give that information in the text.
Line 240- the analysis/results, review the sentence.
If the information in table 10 is from both focus group participants and key informant interviews then it has to be reviewed, because key informants can be counted as individuals but focus group participants are in a group and not all of them might agree on an issue or point.
Don’t report the focus findings in %, instead you can report it qualitatively, in text form e.g., most of the focus group participants agreed that agricultural expansion is one of the direct causes of land cover change.
Line 261- report your findings separately for the focus group discussions and key informant interviews
Line 372- Can you please provide the total rainfall and average temperature for the study area in 1990 and 2022.
Can you provide scientific evidence to show the relationship between the effect of the main climate elements and LU/LC changes?

Reviewer 3 ·

Basic reporting

I appreciate the efforts of the authors to synthesize this manuscript. Overall, the manuscript is concise and generally well organized so that it is easy to read and understand. The Title of the manuscript is short and precise. It brings up important insights about the link between land use/cover change and its driving forces at the local scales. The authors tried to integrate a wide spectrum of interacting environmental variables with spatio-temporal dimensions. It properly highlights the general impression of what work is about. However, the Authors did not demonstrate methodological novelty or at least presents established ideas in a new way & innovative insights. Most of the methodology employed and most of the results reported (for example; the drivers identified, the implications forwarded and the conclusions reached) are already well known and not very novel. Several studies have been carried out in relation to the topic in southern Region of Ethiopia, even in the rift valley area itself. So what is the practical value & the unique contribution this research will make? What is new aspect of this paper that is academically innovative? Please show the uniqueness of this study from theoretical and methodological perspectives by comparing it with existing related studies.3. The literature review needs to be more comprehensive and critical. In this regard, the background of the manuscript does not clearly indicate the theoretical and empirical basis and not supported by the latest publications in the field of study. Most of the literatures cited are a bite outdated and does not show current developments/ contemporary evidence.

Experimental design

The innovation of applied methodologies are not sufficiently described (or inadequate) and the choices for each approaches and techniques are not sufficiently justified. For example, authors have not indicated the reasons for the selection of the initial and others years such as 1990, 2006 & 2022 for the selection of satellite images and for land use and land cover dynamics. Why these period and time interval were considered?
I am very concerned with the accuracy of the overall and intra land use/cover change detection & the validity of maps generated. Just to cite some examples, as a rule of thumb, Congalton (1991) recommends a minimum of 75-100 sample points per LULC categories. However, the authors of this paper do not show how many training samples they took for all LULC classes. Put in other words, how many of the validation points were ground truthed? As geographic phenomena are dynamic, how they did validated maps for historical data using Google Earth?

Validity of the findings

1. The authors did not clearly quantify inter-category transitions such as gain, losses, swapping and persistence of for each LU/LC categories in the landscape. The nature of change is not clearly articulated. Is that random or systematic? If both, which one is the dominant random or systematic LULC change?
2. What the authors presented related to major drivers of LULC Change in study area seem more of assumption/theory! Quantitative results are needed to support qualitative narrations. In this regard, many statements are not verified by statistically sound evidences. For example, Population growth and climate change are attributed to be the drivers of LULCC in the study area. However, a percentage increase in population and quantitative change in climate variables for respective study periods are not statistically shown. I would suggest to fully rewriting the paper with supporting quantitative evidences. Otherwise, it is hard to believe.
3. The conclusions part seems a brief summary and does not explicitly show the implications of the findings. The authors just simply reiterate what they found, not rather discuss what the findings mean in relation to the theoretical body of knowledge on the topic.
4. Avoid improper expression: for example Southern Nations, Nationalities, and Peoples Region (line 108) is recently restructured into various regions and hence, better to use the new name of the region in which the study area is found

Additional comments

• Given the above pitfalls, the manuscript in its current form does not fully qualify for publication. The paper needs minor revisions to bring it to publishable standard.

---

## Round 0.2 · Major Revisions

These comments should be treated and revised carefully. Especially explaining your academic contribution.

Reviewer 1 ·

Basic reporting

the figures quality need improvement. such as, Figure 2, Figure 3

Experimental design

no comment

Validity of the findings

I suggest the author can add the paper's findings compared with others research.

Additional comments

Although the author has revised paper, some questions still exist in the paper. Mainly including, method adopt reasons? "Major drivers of LULC changes" need more deep study. I suggest the author can adopt some statistic method to study the drivers. In addition, there are many tables in the paper, I suggest some unimportant table can put in the supplement files.

---

## Round 0.3 · accepted · Accept

In my opinion, the article can be accepted for publication in PeerJ